# Molecular Detection and Analysis of Blast Resistance Genes in Rice Main Varieties in Jiangsu Province, China

Zhongqiang Qi [1,2], Yan Du [1], Junjie Yu [1], Rongsheng Zhang [1], Mina Yu [1], Huijuan Cao [1], Tianqiao Song [1], Xiayan Pan [1], Dong Liang [1] and Yongfeng Liu [1,2,*]

1 Institute of Plant Protection, Jiangsu Academy of Agricultural Science, Nanjing 210014, China
2 IRRI-JAAS Joint Laboratory, Jiangsu Academy of Agricultural Science, Nanjing 210014, China
* Correspondence: liuyf@jaas.ac.cn

**Abstract:** Rice blast, caused by *Pyricularia oryzae*, is one of the most destructive rice diseases worldwide. Using resistant rice varieties is the most cost-effective way to control the disease, and it is crucial to analyze the resistance level and the resistance genes distribution of the main varieties. In this study, we collected 119 rice main varieties in Jiangsu province and evaluated the resistance to leaf and panicle blast and found that *indica* rice was more resistant to rice blast than *japonica* rice. Moreover, we detected the distribution of 14 resistance genes (*R* genes) in the 119 varieties. The distribution frequencies of three *R* genes, *Pish*, *Pit*, and *Pia*, were higher than 80%, and the *Pigm* had the lowest distribution frequency (1.68%), followed by *Pi2* (15.18%) and *Pi5*, *Piz-t* (24.37%). Combined with the multiple stepwise regression and the resistance contribution rate, eight major *R* genes *Pita*, *Pi5*, *Pi9*, *Pib*, *Pb1*, *Pikm*, *Piz-t*, and *Pi2* significantly affected the resistance of rice, and we also found that six gene combinations with 100% resistance contribution rate could effectively increase the resistance of rice varieties. In summary, monitoring the resistance level of rice varieties and analyzing their resistance genes were beneficial for rice resistance breeding.

**Keywords:** *Pyricularia oryzae*; esistance gene; molecular marker-assisted selection; gene pyramiding; resistance contribution rate

## 1. Introduction

Rice (*Oryza sativa*) is one of the most globally important staple foods which contributes 23% of the calories consumed by 50% of the global human population [1,2]. Rice production is threatened by rice blast disease caused by the ascomycete fungus *Pyricularia oryzae* (formerly *Magnaporthe oryzae*), which causes 10–30% of the world's food loss each year [3–5]. The rice-growing areas in Jiangsu province are located in the south-north climate transition zone, which results the suitable temperature and more precipitation for the occurrence of rice blast, and the average annual occurrence area reaches about 730,000 hectares, which seriously threatens the safe production of rice [6]. The use of resistant rice varieties is the most economical and efficient method to control the disease [7]. However, the resistance of rice main varieties can weaken or be lost after 3–5 years of field cultivation due to the complexity of physiological species composition and the variation in frequency of fungal strains [8–10]. Therefore, it is critical for rice improvement to identify the resistance level to leaf and panicle blast of rice main varieties and analyze the distribution of resistance genes.

In rice–*P. oryzae* interactions, the dynamics of host resistance and pathogenicity of *P. oryzae* are followed by gene for gene theory, where major resistance gene is effective in preventing infection by an isolate containing the corresponding avirulence gene [11,12]. So far, over 100 rice blast resistance genes have been identified, and 25 major *R* genes and 2 partial resistance genes (*Pi21* and *Pb1*) have been cloned and characterized [13–19]. All of the cloned *R* genes (except *Pid2* and *Pi21*) are those of the nucleotide-binding site and leucine-rich repeat (NBS-LRR) class [20]. *Pita* is located on chromosome 12 and encodes

the plasma membrane receptor protein, which is related to the avirulent gene *Avr-Pita* of the *P. oryzae* interaction of expression products which leads to a resistance response [21]. Seven alleles including *Pik*, *Pikp*, *Pikh*, *Pikm*, *Piks*, *Pike*, and *Pi1* were identified in the *Pik* locus, which requires two adjacent NBS-LRR class genes for full functionality [22]. Another resistance gene locus is the *Piz* locus, which contains *Pi9*, *Pi2*, *Piz-t*, *Pigm*, and *Pi50*. *Pigm* encodes two proteins, *PigmR* and *PigmS*, which regulate broad spectrum durable resistance and yield [23].

The conventional breeding method is to introduce resistance genes into the target lines to develop resistant varieties through sexual hybridization, agronomic traits selection, and resistance identification, which is effective, but it has the disadvantages of heavy workload and lon—time consumption. Molecular marker-assisted selection (MAS) is a selective aid method that applies molecular markers in rice breeding. Functional markers (FMs) refer to marker sites representing specific resistance genes, and the rice traits can be selected by screening molecular markers [24]. Some FMs specific to *R* genes, such as *Pib*, *Pigm*, *Pita*, and *Pikm* have been developed, which will provide a convenient way to identify target genes [25–28]. Moreover, some PCR-based tightly linked markers (LMs) were developed, associated with several *R* genes including *Pi2*, *Pi5*, and *Pi9* [29,30]. The LMs provide an efficient and rapid method for screening target *R* genes in gene introgression and gene pyramiding.

As is well known, the resistance spectrum and resistance level of rice varieties were improved with multiple resistance genes pyramiding [31]. Liu et al. [32] reported that *Pita2*, *Pi5*, *Pi9*, and *Piz-t* showed high frequency and resistance to rice blast in 56 rice main varieties in Yunnan province. In addition, the combination patterns "*Pi9 + Pi54*", "*Pid3 + Pigm*", "*Pi5 + Pid3 + Pigm*", and "*Pi5 + Pi54 + Pid3 + Pigm*" in *indica*-type accessions and "*Pi5 + Pib*", "*Pik + Pita*", "*Pik + Pb1*", "*Piz-t + Pia*" and "*Piz-t + Pita*" in *japonica*-type accessions are critical for the resistance of rice varieties [33].

Rice blast can be classified into leaf blast and panicle blast based on the infected organization, where panicle blast is more destructive in terms of yield loss [34,35]. More and more evidence suggests that there are different regulation mechanisms between leaf blast resistance and panicle blast resistance [35]. The above studies are based on the leaf blast resistance. It is necessary to use the resistance composite index of leaf and panicle blast to identify the resistance of rice varieties.

In this study, we evaluated the resistance level of 119 rice main varieties in Jiangsu province, analyzed the distribution of major resistance genes of the above varieties using the molecular markers of 14 resistance genes, and finally obtained the suitable resistance genes and gene combinations of Jiangsu rice-growing regions. The results provide applicable resistance resources for rice resistance breeding.

## 2. Materials and Methods

### 2.1. Collection of Rice Main Varieties

A set of 119 rice main varieties (94 *Japonica* and 25 *Indica*) were collected from the seed management station of Jiangsu province, and were planted in all the growing regions (Northern, Central, and Southern Jiangsu) of Jiangsu province. These varieties have a large planting area and represent the main varieties in Jiangsu province.

### 2.2. Plant Infection Assays

In this study, the leaf blast and panicle blast were naturally induced by a disease nursery in Jintan (31°40′20″ N, 119°21′34″ E). Each variety was planted in 25 (5 × 5) holes, totaling 1 m², one row (5 holes) each, and susceptible checks (Co39) were planted between each variety and along the borders to facilitate the full spread of the disease. The leaf blast severity was assessed 25 days after sowing and continued at 5 days intervals until the fortieth day of sowing or when the susceptible checks had disease symptoms (85%), whichever occurred earlier [36]. Observations on the panicle blast reaction of the varieties were recorded after 140 or 150 days (rice dough grain stage) after sowing.

*2.3. Disease Assessments*

The disease severity of leaf blast was evaluated using the standard 0–9 scale [6] rated on ten levels, defined as follows: level 0: no disease symptoms; level 1: brown spots, diameter ≤ 1 mm; level 2: large brown spots, diameter 1–2 mm; level 3: circular to elliptic gray lesions, diameter 1–2 cm; level 4: spindle spots, length 1–2 cm, confined between two veins, the infected area does not exceed 2% of the leaf area; level 5: spindle spots, the infected area accounts for 2–10% of the leaf area; level 6: spindle spots, the infected area accounts for 11–25% of the leaf area; level 7: spindle spots, the infected area accounts for 26–50% of the leaf area; level 8: spindle spots, the infected area accounts for 51–75% of the leaf area; level 9: spindle spots, the infected area is more than 75% of the leaf area.

The spectrum of disease reaction for panicle blast was scored visually on a 0–9 scale rated on six levels, defined as follows: level 0: no disease symptoms; level 1: individual branch diseased, loss per panicle less than 5%; level 3: about 1/3 of the branch diseased, loss of 6–20% per panicle; level 5: rice panicle or spindle diseased, loss of 21–50% per panicle; level 7: rice panicle diseased, loss of 51–70% per panicle; level 9: loss of 71–100% per panicle.

The standard for grading the incidence of rice panicle blast: level 0: no disease symptoms; level 1: incidence of panicle blast < 5.0%; level 3: incidence of panicle blast 5.1–10.0%; level 5: incidence of panicle blast 10.1–25.0%; level 7: incidence of panicle blast 25.1–50.0%; level 9: incidence of panicle blast 50.1–100.0%. The incidence of panicle blast = (the number of diseased panicles/100 (the number of investigated panicles)) × 100%.

The standard for grading loss index of panicle blast: level 0: no disease symptoms; level 1: loss of panicle < 5.0%; level 3: loss of panicle 5.1–15.0%; level 5: 15.1–30.0%; level 7: loss of panicle 30.1–50.0%; level 9: 50.1–100%.

The standard for grading composite index of rice blast: level 0: <0.1; level 1: 0.1–2.0; level 3: 2.1–4.0; level 5: 4.1–6.0; level 7: 6.1–7.5; level 9: 7.6–9.0. Composite index = level of leaf blast × 25% + level of incidence of panicle blast × 25% + level of loss index of panicle blast × 50%. Taking the rice variety Suxianggeng1785 as an example, the composite index = 2 × 0.25 + 9 × 0.25 + 5 × 0.5 = 5.25, based on the standard, the grading incidence was level 5.

*2.4. Molecular Screening for Rice Blast R Genes*

For molecular screening, the rice varieties were genotyped for the presence of 14 major blast resistance genes *Pit*, *Pish*, *Pib*, *Pi1*, *Pia*, *Pi54*, *Pita*, *Pi9*, *Pi2*, *Pikm*, *Pigm*, *Pi5*, *Pb1*, and *Piz-t*. A total of 14 molecular markers were selected from published primer sequences and used for molecular screening. The details of information of the primer pairs were provided in Table S1 [37–48].

PCR amplification was carried out in a final volume of 20 μL: 10 μL 2 × Phanta Flash Master Mix (Vazyme, Nanjing, China), 1 μL (20 ng) of template DNA, 1 μL of each primer and 7 μL ddH$_2$O. The PCR program was set up as follows: initial denaturation at 95 °C for 5 min; 35 cycles of 95 °C for 30 s, 58 °C for 30 s, and 72 °C for 30 s; followed by 72 °C for 10 min [49]. The PCR products were detected on the 1.5% agarose gel electrophoresis running at 80 V for 90 min. The gel images were photographed. The PCR amplified fragments were scored as presence (1) or absence (0).

*2.5. Data Analysis*

For diversity analysis, the binary matrix representing fourteen resistance genes which were scored as binary data whether presence (1) and absence (0) was used for estimation of genetic distance and similarity coefficients [38]. The data matrix was further analyzed using NTSYS-pc v.2.1 (Exeter Software, Setauket, NY, USA). The SIMQUAL program in NTSYS-pc v.2.1 software was used to calculate Jaccard's similarity coefficients. The resulting similarity matrix was used for the unweighted pair group method with arithmetic mean (UPGMA)-based dendrogram construction [50].

The relationship between *R* genes and resistance level was analyzed with a multiple stepwise regression using SPSS 20 (SPSS Inc., Chicago, IL, USA). The mean composite index of each variety over three years was used as the resistance level.

## 3. Results

### 3.1. Genotypic Assays for 14 R Genes in the Rice Main Varieties

SSR molecular markers were used to detect the resistance genes identification. Gel electrophoresis patterns of some varieties for 14 *R* genes markers are shown in Figure 1. Seven *R* genes including *Pit*, *Pish*, *Pib*, *Pita*, *Pb1*, and *Pi9* were easily identified based on their molecular markers. Since the molecular markers of the other seven genes, including *Pi54*, *Pi5*, *Pi2*, *Pi1*, *Pikm*, *Pia*, *Piz-t/Pi2*, and *Pigm*, can amplify 2–3 bands, the identification of the above genes was based on specific electrophoresis bands (Figure 1).

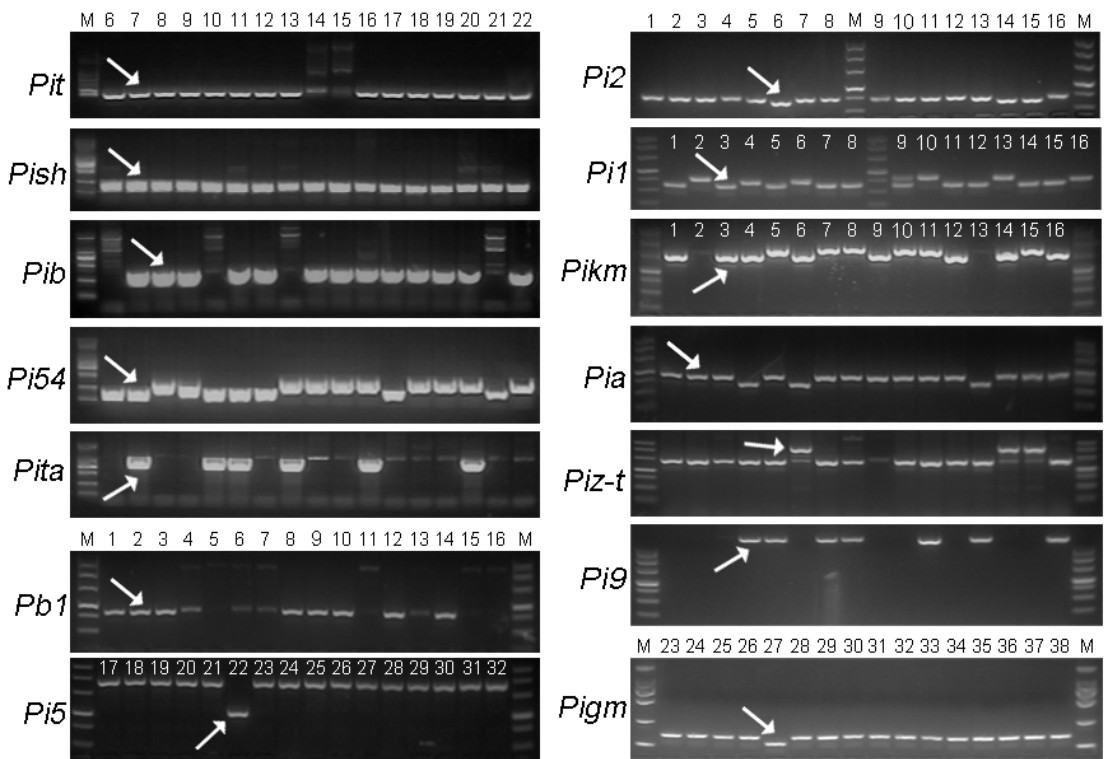

**Figure 1.** PCR amplification patterns of molecular markers that discriminate 14 *R* genes. The arrowheads indicate positive stripe. The numbers were listed in Table S2.

### 3.2. Phenotyping of Rice Blast Disease

According to the screening composite index scores against leaf and panicle blast (Table S2), twenty-three (92.00%) and twenty-eight (29.79%) varieties were resistant and moderately resistant (level 1 and level 3) in *indica* and *japonica* rice, respectively, while fifty (53.19%) varieties were moderately susceptible (Figure 2A). These results indicated that the resistance of *indica* rice to rice blast was better than *japonica* rice and the risk of resistance degradation was greater in *japonica* rice. Moreover, four *japonica* rice varieties (Suxianggeng100, Jinxiangyu1, Jia58 and HuLPR18) and four *indica* rice varieties (Longliangyou1307, Heliangyou332, Huiliangyou882, and Liangyou688) showed better resistance (Table S3). In addition, all the 119 rice varieties were classified into three rice-growing regions such as Northern (74), Central (21), and Southern (24) Jiangsu (Table 1 and Table S2). In total, 50.00% of rice varieties in Southern Jiangsu were resistant and moderately resistant, which was higher than 44.60% in Northern Jiangsu and 28.57% in Central Jiangsu (Table 1), which suggested that there were significant differences in resistance to rice blast of rice varieties in different growing regions.

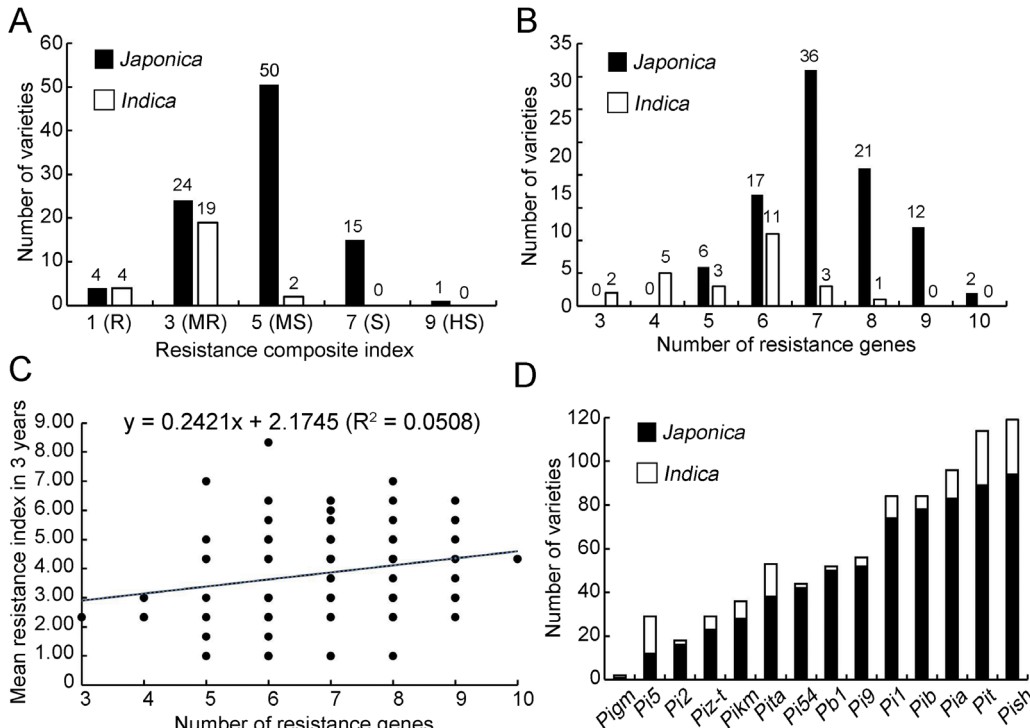

**Figure 2.** The resistance level and 14 *R* genes detection of 119 rice main varieties. (**A**) The resistance level of 119 rice main varieties. (**B**) The number of *R* genes present in 119 rice main varieties. (**C**) The relationship between the number of *R* genes and resistance level. (**D**) The distribution of 14 *R* genes in 119 rice main varieties.

**Table 1.** The resistance composite index of 119 rice main varieties in three regions of Jiangsu province.

| Rice-Growing Regions | | Northern Jiangsu | | Central Jiangsu | | Southern Jiangsu | |
|---|---|---|---|---|---|---|---|
| | | Number of Varieties | Distribution Frequency (%) | Number of Varieties | Distribution Frequency (%) | Number of Varieties | Distribution Frequency (%) |
| Resistance composite index | 1 (R) | 4 | 5.41% | 1 | 4.76% | 3 | 12.50% |
| | 3 (MR) | 29 | 39.19% | 5 | 23.81% | 9 | 37.50% |
| | 5 (MS) | 29 | 39.19% | 12 | 57.14% | 11 | 45.83% |
| | 7 (S) | 12 | 16.22% | 3 | 14.29% | 0 | 0.00% |
| | 9 (HS) | 0 | 0.00% | 0 | 0.00% | 1 | 4.17% |
| Total | | 74 | 100.00% | 21 | 100.00% | 24 | 100.00% |

### 3.3. The Resistance Level Is Positively Correlated with the Resistance Genes

The number of *R* genes detected in the rice main varieties was normally distributed. The frequency of the positive allele of *R* gene ranges from three genes (2 rice varieties) to ten genes (2 rice varieties) in the 119 varieties, mostly ranging from 6–9 genes in *japonica* rice and 4 or 6 genes in *indica* rice (Figure 2B). In addition, correlation analysis showed that the resistance level was weakly positively correlated with the number of resistance genes ($R^2 = 0.0508$, $p < 0.05$) (Figure 2C).

A total of 14 *R* genes were detected in 119 rice main varieties using 14 SSR molecular markers and all the *R* genes were present in both the *indica* and *japoni*ca sub-species genomes (Figure 2D). The distribution frequency of *Pish* was highest (100%), but that of *Pigm* was lowest and detected in only two varieties (Jinxiangyu1 and Wandao153) (Table S2). In addition, although the number of *japonica* rice was much higher (94/25) than that of *indica* rice, the detection rate of *Pi5* was higher (17/12) in *indica* than in *japonica* rice, but the distribution frequency of other *R* genes in *japonica* was higher than that of in *indica* rice

(Figure 2D and Table 2). The distribution characteristics of the *R* genes imply that the *R* gene conferring resistance to *P. oryzae* may differ between the *indica* and *japonica* rice.

**Table 2.** The distribution frequency and resistance contribution rate of 14 *R* genes.

| Resistance Genes | | *Pit* [a] | *Pish* [a] | *Pib* [a] | *Pi54* | *Pita* [b] | *Pi9* | *Pi2* [b] | *Pi1* [a] | *Pikm* [b] | *Pigm* [b] | *Pia* [a] | *Pi5* [b] | *Pb1* | *Piz-t* [b] |
|---|---|---|---|---|---|---|---|---|---|---|---|---|---|---|---|
| **Number of varieties** | | 114 | 119 | 84 | 44 | 53 | 56 | 18 | 84 | 36 | 2 | 96 | 29 | 52 | 29 |
| Distribution frequency (%) | | 95.80 | 100 | 70.59 | 36.97 | 44.54 | 47.06 | 15.13 | 70.59 | 30.25 | 1.68 | 80.67 | 24.37 | 43.70 | 24.37 |
| Resistance composite index | 1 I | 7 | 8 | 4 | 0 | 5 | 1 | 2 | 4 | 4 | 1 | 6 | 2 | 1 | 6 |
| | 3 (MR) | 39 | 43 | 24 | 11 | 26 | 14 | 9 | 26 | 17 | 1 | 30 | 17 | 13 | 9 |
| | 5 (MS) | 52 | 52 | 42 | 24 | 18 | 29 | 6 | 42 | 13 | 0 | 46 | 7 | 27 | 12 |
| | 7 (S) | 15 | 15 | 13 | 8 | 4 | 11 | 1 | 11 | 2 | 0 | 14 | 3 | 11 | 2 |
| | 9 (HS) | 1 | 1 | 1 | 1 | 0 | 1 | 0 | 1 | 0 | 0 | 0 | 0 | 0 | 0 |
| Resistance contribution rate (%) | | 40.35 | 42.86 | 33.33 | 25.00 | 58.49 | 26.79 | 61.11 | 35.71 | 58.33 | 100 | 37.50 | 65.52 | 26.92 | 51.72 |

Note: "a" indicates the *R* genes with the distribution frequency higher than 70%; "b" indicates the *R* genes with the resistance contribution rate higher than 50%.

### 3.4. Cluster Analysis of the 119 Rice Main Varieties

The 119 rice main varieties were categorized into two clusters (I and II) at 60% level of genetic similarity coefficient (Figure 3). Major cluster I which contained 107 varieties, was divided into two sub-clusters IA and IB. Moreover, sub-cluster IA consisting of 78 varieties was divided into two sub-clusters IA-1 and IA-2. IA-1 contained 64 varieties, of which 13 (20.31%) are resistant (varieties resistant and moderately resistant). IA-2 consists of 14 varieties, with 7 resistant genotypes (50.00%). Similarly, sub-cluster IB included 29 varieties which were divided into two sub-clusters, IB-1 and IB-2, and most *indica* rice varieties (21/25) were clustered in these groups. Sub-cluster IB-1 consisted of 9 varieties, of which, all the varieties are resistant (100%). Further, sub-cluster IB-2 contained 20 varieties, of which 15 (75.00%) are resistant. Interestingly, the rice varieties of sub-cluster IB were mostly (23/29) grown in Northern Jiangsu, but that of other clusters were grown in Northern, Central and Southern Jiangsu. On the other hand, cluster II was divided into two sub-clusters IIA (1) and IIB (11) and 7 varieties (58.33%) were resistant. These results indicated that sub-cluster IB contained most *indica* rice varieties, which showed better resistance. In addition, we analyzed the distribution of rice varieties contained *R* genes, most *R* genes were relatively consistent regionally, but the rice varieties containing *Pi9*, *Pi5*, and *Piz-t* were mostly grown in Northern Jiangsu, while the rice varieties containing *Pib*, *Pi1*, and *Pia* were mostly found in Central and Southern Jiangsu (Table S4). Some rice varieties of similar ecologies belong to the same cluster, and the genetically similar genotypes of each cluster were characterized by varieties of different ecologies.

### 3.5. Analysis of R Genes Contribution Rate and R Gene Combination Patterns

We analyzed the distribution of 14 *R* genes in 119 rice main varieties using 16 SSR markers (Table 2). The distribution frequency of *Pish* was the highest (100%), followed by *Pit* (95.80%) and *Pia* (80.67%). The *Pigm* gene had the lowest distribution frequency (1.68%), followed by *Pi2* (15.18%) and *Pi5*, *Piz-t* (24.37%). In addition, based on the resistance composite index, we calculated the resistance contribution rates of each *R* gene. Interestingly, the resistance contribution rate of *Pigm* was the highest (100%), because *Pigm* was detected in only two varieties (Jinxiangyu1 and Wandao153) (Table S2), which showed resistance and moderately resistant. The resistance contribution rates of *Pi5*, *Pi2*, *Pita*, *Pikm*, and *Piz-t* were 65.52%, 61.11%, 58.49%, 58.23%, and 51.72%, respectively, which were all greater than 50%.

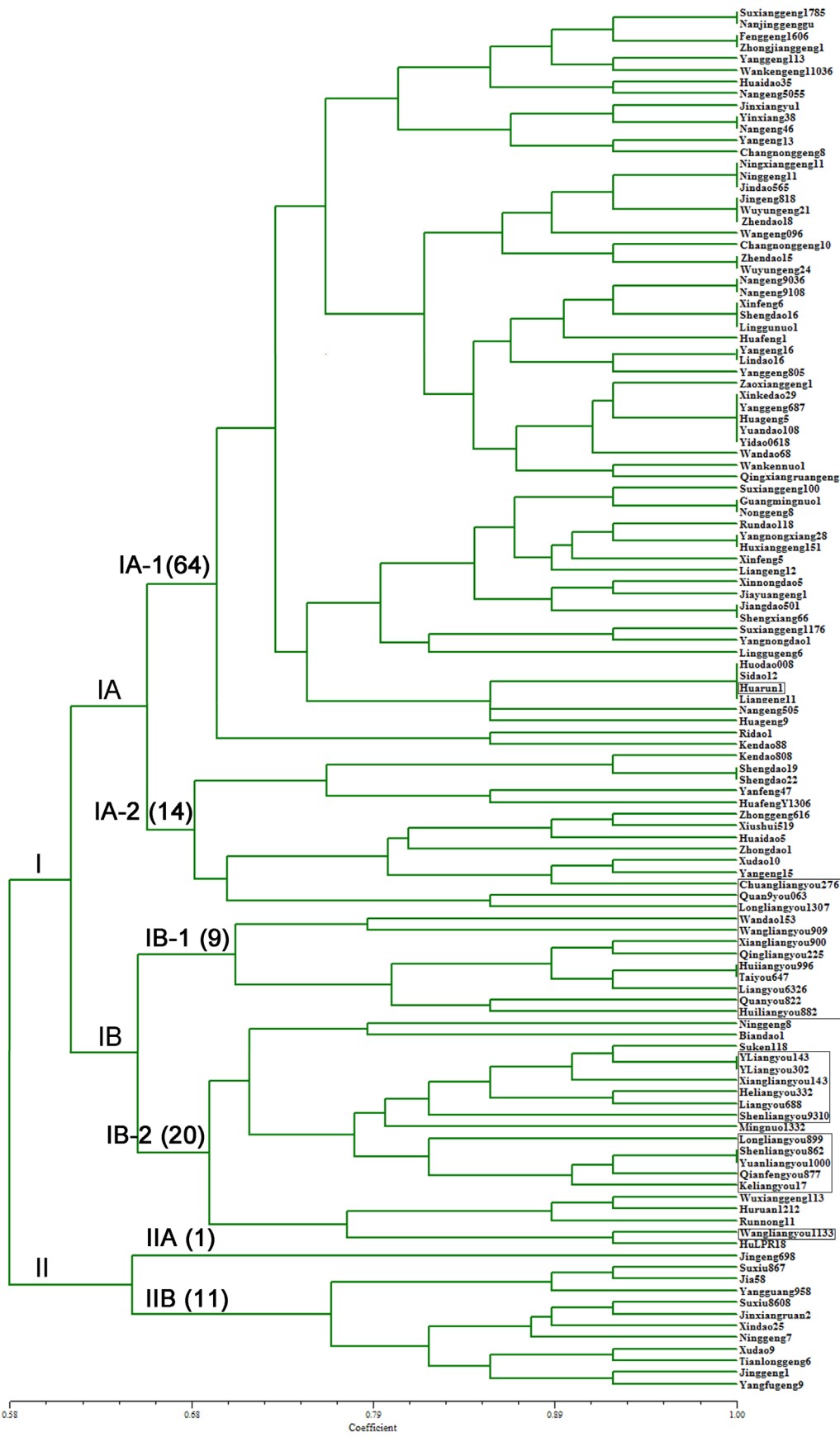

**Figure 3.** Clustered analysis of 119 rice main varieties on 14 *R* genes.

To explore which *R* genes affect rice resistance, we analyzed the relationship between the resistance level and *R* genes using multiple stepwise regression. By fitting this model, seven variables (*Pi9*, *Pib*, *Pit*, *Pita*, *Pi5*, *Pigm*, and *Pb1*) were entered (Table 3) and we found that model 7 including the seven *R* genes was the best ($R^2$ = 0.434, Sig = 0.000) (Tables 4 and S5). These results indicated that *Pi9*, *Pib*, *Pit*, *Pita*, *Pi5*, *Pigm*, and *Pb1* significantly affected the resistance of rice. Combined with the resistance contribution rate data above, we analyzed the resistance contribution rate of the gene polymerization models (Table 5). A total of 16 combinations including three genes and six combinations including four genes were found and contained eight *R* genes *Pita*, *Pi5*, *Pi9*, *Pib*, *Pb1*, *Pikm*, *Piz-t* and *Pi2* (Table 5). The distribution frequency of 22 gene combinations was low, ranging from 0.84% (*Pita* + *Pikm* + *Pi5* + *Piz-t*, *Pita* + *Pikm* + *Pi2* + *Piz-t*) to 16.81% (*Pi9* + *Pib* + *Pb1*). In contrast, the resistance contribution rate of these gene combinations was high, ranging from 16.67% to 100%. Among them, there were six combinations with 100% and one combination with 80% resistance contribution rate, which were *Pita* + *Pikm* + *Piz-t*, *Pita* + *Pikm* + *Pi2*, *Pita* + *Pi5* + *Piz-t*, *Pikm* + *Piz-t* + *Pi2*, *Pita* + *Pikm* + *Pi5* + *Piz-t*, and *Pita* + *Pikm* + *Pi2* + *Piz-t*, *Pita* + *Pikm* + *Pi5* (Tables 5 and S6). It is worth noting that the number of varieties containing these gene combinations is small, with only the number of varieties in combination *Pikm* + *Piz-t* + *Pi2* reaching five and the remaining five combinations only 1–2 varieties. These results suggested that there were some key *R* genes or gene combinations for rice resistance, which had great potential for utilization in Jiangsu province.

**Table 3.** Seven variables were entered with multiple stepwise regression.

| Variables Entered/Removed | | | |
|---|---|---|---|
| **Model** | **Variables Entered** | **Variables Removed** | **Method** |
| 1 | *Pi9* | . | Stepwise (Criteria: Probability of F-to-enter $\leq$ 0.050, Probability of F-to-remove $\geq$ 0.100) |
| 2 | *Pib* | . | Stepwise (Criteria: Probability of F-to-enter $\leq$ 0.050, Probability of F-to-remove $\geq$ 0.100) |
| 3 | *Pit* | . | Stepwise (Criteria: Probability of F-to-enter $\leq$ 0.050, Probability of F-to-remove $\geq$ 0.100) |
| 4 | *Pita* | . | Stepwise (Criteria: Probability of F-to-enter $\leq$ 0.050, Probability of F-to-remove $\geq$ 0.100) |
| 5 | *Pi5* | . | Stepwise (Criteria: Probability of F-to-enter $\leq$ 0.050, Probability of F-to-remove $\geq$ 0.100) |
| 6 | *Pigm* | . | Stepwise (Criteria: Probability of F-to-enter $\leq$ 0.050, Probability of F-to-remove $\geq$ 0.100) |
| 7 | *Pb1* | . | Stepwise (Criteria: Probability of F-to-enter $\leq$ 0.050, Probability of F-to-remove $\geq$ 0.100) |

**Table 4.** The analysis of seven model summary.

| Model Summary | | | | |
|---|---|---|---|---|
| **Model** | **R** | **R Square** | **Adjusted R Square** | **Std. Error of the Estimate** |
| 1 | 0.340 [a] | 0.116 | 0.108 | 1.40538 |
| 2 | 0.453 [b] | 0.206 | 0.192 | 1.33772 |
| 3 | 0.512 [c] | 0.263 | 0.243 | 1.29445 |
| 4 | 0.561 [d] | 0.314 | 0.29 | 1.25364 |
| 5 | 0.606 [e] | 0.367 | 0.339 | 1.20997 |
| 6 | 0.636 [f] | 0.405 | 0.373 | 1.1781 |
| 7 | 0.659 [g] | 0.434 | 0.398 | 1.15441 |

Note: [a]. Predictors (Constant): *Pi9*; [b]. Predictors (Constant): *Pi9*, *Pib*; [c]. Predictors (Constant): *Pi9*, *Pib*, *Pit*; [d]. Predictors (Constant): *Pi9*, *Pib*, *Pit*, *Pita*; [e]. Predictors (Constant): *Pi9*, *Pib*, *Pit*, *Pita*, *Pi5*; [f]. Predictors (Constant): *Pi9*, *Pib*, *Pit*, *Pita*, *Pi5*, *Pigm*; [g]. Predictors (Constant): *Pi9*, *Pib*, *Pit*, *Pita*, *Pi5*, *Pigm*, *Pb1*.

**Table 5.** The distribution frequency and resistance contribution rate of different gene combinations.

| Polymerization Model | Number of Varieties | Distribution Frequency (%) | Variety Number whose Composite Resistance Index ≤3 | Variety Number Whose Composite Resistance Index >3 | Resistance Contribution Rate (%) |
|---|---|---|---|---|---|
| *Pita + Pi5 + Pi9* | 6 | 5.04 | 4 | 2 | 66.67 |
| *Pita + Pi5 + Pib* | 4 | 3.36 | 2 | 2 | 50.00 |
| *Pita + Pi5 + Pb1* | 5 | 4.20 | 3 | 2 | 60.00 |
| *Pita + Pi9 + Pib* | 18 | 15.13 | 8 | 10 | 44.44 |
| *Pita + Pi9 + Pib* | 14 | 11.76 | 5 | 9 | 35.71 |
| *Pita + Pib + Pb1* | 15 | 12.61 | 6 | 9 | 40.00 |
| *Pi5 + Pi9 + Pib* | 8 | 6.72 | 3 | 5 | 37.50 |
| *Pi5 + Pi9 + Pb1* | 11 | 9.24 | 2 | 9 | 18.18 |
| *Pi5 + Pib + Pb1* | 6 | 5.04 | 1 | 5 | 16.67 |
| *Pi9 + Pib + Pb1* | 20 | 16.81 | 4 | 16 | 20.00 |
| *Pita + Pikm + Pi5* | 5 | 4.20 | 4 | 1 | 80.00 |
| *Pita + Pikm + Piz-t* | 2 | 1.68 | 2 | 0 | 100.00 |
| *Pita + Pikm + Pi2* | 2 | 1.68 | 2 | 0 | 100.00 |
| *Pita + Pi5 + Piz-t* | 2 | 1.68 | 2 | 0 | 100.00 |
| *Pita + Piz-t + Pi2* | 4 | 3.36 | 2 | 2 | 50.00 |
| *Pikm + Piz-t + Pi2* | 5 | 4.20 | 5 | 0 | 100.00 |
| *Pita + Pi5 + Pi9 + Pib* | 4 | 3.36 | 2 | 2 | 50.00 |
| *Pita + Pi5 + Pi9 + Pb1* | 4 | 3.36 | 2 | 2 | 50.00 |
| *Pita + Pi5 + Pib + Pb1* | 3 | 2.52 | 1 | 2 | 33.33 |
| *Pib + Pi5 + Pi9 + Pb1* | 6 | 5.04 | 1 | 5 | 16.67 |
| *Pita + Pikm + Pi5 + Piz-t* | 1 | 0.84 | 1 | 0 | 100.00 |
| *Pita + Pikm + Pi2 + Piz-t* | 1 | 0.84 | 1 | 0 | 100.00 |

## 4. Discussion

Breeding and utilization of resistant rice varieties is the most economical and effective measure to control the occurrence and epidemic of rice blast [51]. It is necessary to analyze the distribution of resistance genes and resistance level of rice main varieties in Jiangsu growing region, which also was of great significance for accurate utilization of *R* genes. In this research, we monitored the resistance level of 119 rice main varieties in Jiangsu province from 2019 to 2021 using the composite index scores against leaf and panicle blast. Moreover, the distribution frequencies of 14 *R* genes were detected with the SSR molecular markers.

*P. oryzae* could infect the whole growth period of rice, mainly leaf and panicle and panicle blast is considered to be a more serious phase of this disease [52]. Previous studies have reported that there is a certain correlation between the resistance to leaf blast and panicle blast, but there are also inconsistent cases of resistance [53,54]. Moreover, the resistance level of major *R* genes in rice was not consistent in different growth stages. Some *R* genes with resistance to leaf blast did not effect panicle blast [35]. Therefore, the comprehensive evaluation of blast resistance by leaf blast and panicle blast is an important technical means for rice resistance breeding and evaluation of cultivar resistance. In this study, we evaluated the comprehensive resistance level of 119 rice main cultivars to leaf and panicle blast in 2019, 2020, and 2021, these data more objectively and accurately reflected the resistance of 119 varieties to rice blast.

There were obvious differences in the resistance level of different varieties (lines) to rice blast. Based on the analysis of the composite index of 119 rice main varieties against leaf and panicle blast, the resistance level of *indica* rice to rice blast was better than *japonica* rice (Figure 2A and Table S2). This result may be related to the cultivar types in Jiangsu province. The area of *japonica* rice planted in Jiangsu province is 1,933,000 hectares, nearly 87.9% of the total area [55], the *P. oryzae* population in Jiangsu province was more suitable for *japonica* rice varieties under the selection pressure of *japonica* rice for a long time. In other words, the pathogenicity of the *P. oryzae* to *japonica* rice varieties was stronger than that of *indica* rice varieties. Hao et al. [56] reported that 47.5% of 800 *indica* rice varieties in rice areas in the middle and lower reaches of the Yangtze River (*indica* rice cultivation area) showed moderate susceptibility to rice blast, while only 0.2% showed resistance.

The present study categorized 119 rice main varieties into two groups at 60% level of genetic similarity and further subdivided them into six sub-clusters: IA-1, IA-2, IB-1, IB-2, IIA, and IIB. Sub-cluster IB contained 21 (84%) *indica* rice varieties, which were grown in northern Jiangsu. By comparison, the other four sub-clusters consisted of almost all *japonica* rice varieties, which were grown in various rice-growing regions in Jiangsu province. These results showed that the genetically similar genotypes of each cluster were differentiated by rice varieties in different ecologies.

So far, more than 500 quantitative trait loci (QTLs) are associated with blast resistance, while about 100 resistance genes or alleles have been identified [57]. Marker-assisted selection (MAS) is a classical tool in rice breeding for improved resistance to rice blast and several functional markers of *R* genes have been developed [58]. In this study, genetic frequencies of 14 *R* genes, *Pigm*, *Pi2*, *Pi5*, *Piz-t*, *Pikm*, *Pi54*, *Pb1*, *Pita*, *Pi9*, *Pib*, *Pi1*, *Pia*, *Pit*, and *Pish* varied from 1.68 to 100% (Table 2). The distribution frequencies of *Pish*, *Pit*, and *Pia* were higher than that of others, but the resistance contribution rates of these three *R* genes were lower than 42.86% (Table 2), in other words, the utilization potential of these genes is not high. Accordingly, the resistance contribution rate of *Pi5*, *Pi2*, *Pita*, *Pikm*, and *Piz-t* were more than 50%, especially *Pigm*, which reached 100%. *Pigm* was identified as a novel broad-spectrum resistance locus [4], but this gene was detected in two varieties, which indicated that *Pigm* showed a higher level of blast resistance, but it had not been widely used. The *Pi5* gene, which clusters with *Pii* on chromosome 9, was a locus associated with broad-spectrum resistance to diverse blast isolates [46]. Moreover, we also found that *Pi5* had a higher distribution rate in *indica* rice than in *japonica* rice (Figure 2D), which may be one reason for the better resistance level to rice blast of *indica* rice than *japonica* rice. He et al. [48] reported that *Pita* was the most widely distributed among the main cultivars in Zhejiang province, accounting for 47.83%, the distribution frequencies of resistance genes *Pi2/Piz-t* and *Pikm* were equal in cultivars (34.78%). In Liaoning and Tianjin, *Pikh* had the widest distribution frequency in 107 main cultivars, reaching 82%, the distribution frequencies of *Pita*, *Pikm*, *Pi5*, and *Pi9* with strong broad-spectrum resistance were relatively less, 38%, 32%, 13%, and 5%, respectively [44]. These results suggested that is different core effective resistance genes (with high distribution frequency) of rice main varieties in different rice planting regions, and the core effective resistance genes of rice varieties in one region are less distributed in rice in another region.

Multiple resistance gene pyramiding is not a simple accumulation of the resistance spectrum of single *R* genes, but a very significant interaction effect among resistance genes [59]. The premise of polygenic breeding is to understand the genetic background of resistance resources and identify the gene combinations [47]. In this study, we found that the resistance level of rice variety is weakly positively correlated with the number of resistance genes (Figure 2C), which is similar to that of Xiao et al. [33]. Therefore, it is not that the more resistance genes pyramided, the better the resistance level of rice variety. The best strategy is to aggregate effective resistance genes. Combined with the multiple stepwise regression and the resistance contribution rate data, 22 gene combinations including three genes (16 combinations) and four genes (6 combinations) and contained eight major *R* genes *Pita*, *Pi5*, *Pi9*, *Pib*, *Pb1*, *Pikm*, *Piz-t*, and *Pi2*. The resistance contribution rates of six gene combinations were 100%, but the number of rice varieties containing these gene combinations is small, which demonstrated that these gene combinations are not widely used in rice breeding. Sun et al. [47] reported that the gene combination patterns *Pi54 + Pib + Pb1*, *Pita + Pib + Pikm*, *Pita + Pib + Pikm + Piz-t*, and *Pita + Pi54 + Pib + Pb1* were the main factors determining the resistance of rice varieties in Jiangsu province. However, the research was mainly based on some rice intermediate materials, not the rice main varieties, and the resistance index used is less than or equal to 5, while in our data, it is less than or equal to 3, the resistance index is more strict, so the gene combination patterns in our study are more accurate.

## 5. Conclusions

In this research, we comprehensively evaluated the resistance to leaf and panicle blast of 119 rice main varieties in Jiangsu province and analyzed the distribution of 14 resistance genes. Moreover, we identified the major resistance genes and gene combination patterns suitable for rice varieties in Jiangsu province by using gene resistance contribution rate and multiple stepwise regression analysis. Our study demonstrates that major effective resistance gene pyramiding can effectively increase the resistance of rice varieties. Moreover, the distribution of resistance genes in rice varieties, the exploitation and utilization of new resistance genes and the monitoring of avirulence genes of blast fungus can better achieve the broad-spectrum durable resistance of rice.

**Supplementary Materials:** The following supporting information can be downloaded at: https://www.mdpi.com/article/10.3390/agronomy13010157/s1, Table S1: List of markers used for blast resistance genes; Table S2: The distribution of 14 resistance genes in 119 rice main varieties in Jiangus province; Table S3: The resistant evaluation of 119 main rice varieties to rice blast from 2019 to 2021; Table S4: The distribution rate of rice varieties containing R genes in three regions of Jiangsu province; Table S5: The ANOVA analysis of seven models; Table S6: The resistant evaluation of rice varieties in seven gene combinations.

**Author Contributions:** Writing—first draft preparation, Z.Q. and Y.D.; writing—review and editing, Z.Q., J.Y., R.Z., M.Y., H.C., T.S., X.P., D.L. and Y.L.; funding acquisition, Y.L. and Z.Q.; figures and tables, Z.Q. All authors have read and agreed to the published version of the manuscript.

**Funding:** This work was supported by funding to Y.L. (Yongfeng Liu), form the Jiangsu Agriculture Science and technology Innovation Fundation grant CX19(1008); This work also received funding from National Natural Science Foundation of China grants 31861143011 (Yongfneg Liu) and 31871921 (Zhongqiang Qi) and Jiangsu Modern Agricultural Technology System of Rice and Wheat Industry JAST (2022) 273 (Yongfeng Liu).

**Data Availability Statement:** Not applicable.

**Conflicts of Interest:** The authors declare no conflict of interest.

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
