# Peer review of "Molecular Detection and Analysis of Blast Resistance Genes in Rice Main Varieties in Jiangsu Province, China"

_agronomy, doi:10.3390/agronomy13010157_

Round 1

Reviewer 1 Report

The manuscript was focused on molecular detection and analysis of blast resistance genes in main rice varieties in Jiangsu province, China. Totally, the manuscript was well organized and easy to follow. The methods and the results used in this study are scientifically sound. Their findings will greatly contribute to control rice blast in the field and are beneficial for rice resistance breeding.

Here, this manuscript has some minor concerns which need to be improved. 

1. In Line 17, the full name of R should be used firstly. Please correct the same problem for the rest.

2. In Lines 19-25 and 28, the font size is inconsistent with others. Correct them.

3. In Line 24, analyzing of resistance genes of rice were beneficial for resistance breeding should be revised as analyzing of their resistance genes were beneficial for rice resistance breeding.

4. In lines 36-37, the sentence is obscure for understanding.

5. In Line 42,  rice main varieties should be revised as  main rice varietiesor main varieties of rice. Please check out the same problem for the rest.

6. In Line 64, whats meaning for screened by screening ?

7. For 2.1 Collection of rice main varieties, I suggested the authors should add a schematic diagram for introducing the distribution of 119 main rice varieties in Jiangsu province.

8. In Lines 96, 97 and 99, of sowing and after 140 or 150 days (Rice dough grain stage) of sowing should be revised as after sowing and 140 or 150 days (rice dough grain stage) after sowing, respectively.

9. In Line 132, why is this Table S5, but not Table S1? Usually, it should be present in an ascending order.

10. In Line 137-138, the sentence should be revised as The PCR products were detected on the 1.5% agarose gel electrophoresis running at 80 V for 90 min. 

11. In Lines 143-144, subject to means a bad encounter, so it is not appropriate. The sentence can be changed into The data matrix was further analyzed using NTSYS-pc v.2.1.

12. In Lines 144 and 149, please provide the developer information for NTSYS-pc v.2.1, SIMQUAL and SPSS 20, respectively.

13. As for Tables 1, 2 and 5, one word or a model name in the first column should be shown reasonably in line break.

14. As for the note of Table S3, a, b, c...g should be corrected as 1, 2, 3, ...7, respectively.

15. As for Table S5, Chr. and R should be written in the full name. 

Author Response

The manuscript was focused on molecular detection and analysis of blast resistance genes in main rice varieties in Jiangsu province, China. Totally, the manuscript was well organized and easy to follow. The methods and the results used in this study are scientifically sound. Their findings will greatly contribute to control rice blast in the field and are beneficial for rice resistance breeding.

Here, this manuscript has some minor concerns which need to be improved. 

  1. In Line 17, the full name of “R” should be used firstly. Please correct the same problem for the rest.

Response: We thank the reviewer for the positive comments. We have modified the full name “Resistance genes” in line 18.

  1. In Lines 19-25 and 28, the font size is inconsistent with others. Correct them.

Response: We have modified the font size of this section.

  1. In Line 24, “analyzing of resistance genes of rice were beneficial for resistance breeding” should be revised as “analyzing of their resistance genes were beneficial for rice resistance breeding”.

Response: Thanks. We have made correction.

  1. In lines 36-37, the sentence is obscure for understanding.

Response: We agree with the reviewer’s point and we have modified with “The rice-growing areas in Jiangsu province are located in the south-north climate transition zone, which results the suitable temperature and more precipitation for the occurrence of rice blast, and the average annual occurrence area reaches about 11 million mu, which seriously threatens the safe production of rice” (line 36-39).

  1. In Line 42, “ rice main varieties” should be revised as “ main rice varieties”or “main varieties of rice”. Please check out the same problem for the rest.

Response: We thank the reviewer for the positive comments and we have replaced with “rice main varieties” in the full text.

  1. In Line 64, what’s meaning for “screened by screening” ?

Response: We have modified with “the rice traits can be selected by screening molecular markers” (line 66)

  1. For “2.1 Collection of rice main varieties”, I suggested the authors should add a schematic diagram for introducing the distribution of 119 main rice varieties in Jiangsu province”.

Response: We thank the reviewer for the critical comment. However, the rice main varieties were collected from the seed management station of Jiangsu province, and there was no geographical information of the varieties. Therefore, we classified the varieties into Northern Jiangsu, Central Jiangsu and Southern Jiangsu according to the suitable planting areas in Table S2.

  1. In Lines 96, 97 and 99, “of sowing” and “after 140 or 150 days (Rice dough grain stage) of sowing” should be revised as “after sowing” and “140 or 150 days (rice dough grain stage) after sowing”, respectively.

Response: Thanks. We have made correction.

  1. In Line 132, why is this Table S5, but not Table S1? Usually, it should be present in an ascending order.

Response: We thank the reviewer for the positive comments. We have made modification. Changed Table S5 to Table S1, Table S1 to Table S3, Table S3 to Table S4, Table S4 to Table S5.

  1. In Line 137-138, the sentence should be revised as “The PCR products were detected on the 1.5% agarose gel electrophoresis running at 80 V for 90 min.” 

Response: Thanks. We have made correction.

  1. In Lines 143-144, “subject to” means a bad encounter, so it is not appropriate. The sentence can be changed into “The data matrix was further analyzed using NTSYS-pc v.2.1.”

Response: Thanks. We have made correction.

  1. In Lines 144 and 149, please provide the developer information for “NTSYS-pc v.2.1”, “SIMQUAL” and “SPSS 20”, respectively.

Response: We have added the developer information in line 154, 155 and 159.

  1. As for Tables 1, 2 and 5, one word or a model name in the first column should be shown reasonably in line break.

Response: We thank the reviewer for the positive comments and we have modified the format of Table 1, 2, 5 and others.

  1. As for the note of Table S3, “a, b, c...g” should be corrected as “1, 2, 3, ...7”, respectively.

Response: Thanks. We have made correction.

  1. As for Table S5, “Chr.” and “R” should be written in the full name. 

Response: Thanks. We have made correction.

Reviewer 2 Report

Review of Molecular detection and analysis of blast resistance genes in main rice varieties in Jiangsu province, China, submitted for possible publication in Agronomy.

Comments and Suggestions for Authors

The article is well done and provides interesting information. In general, this article is well written, but some polishing of the English text is suggested. Some suggestions/comments for improving the manuscript are provided below.

35 – 36 - Rice blast is prevalent in rice-growing areas in Jiangsu province due to the climate transition zone between north and south… What are the climatic differences between north and south that make blast prevalent?

71 - … Liu et al … add a dot after al. and the reference number. Check that all other references are correctly cited.

76 - Rice blast can be classified into … this part should go in a separate paragraph.

89 - A set of 119 rice main … main varieties of Jiangsu province?

110 - The spectrum of disease reaction … add that it is for panicle blast. This is severity of panicle blast.

115 – Standard for grading the incidence of rice panicle blast: … What degree of severity was used for incidence evaluation, or was any symptom or degree of severity used to calculate incidence? According with Table S1 100 panicles were evaluated.

124 - Where does this Composite Index formula come from? If it was created here, it should be explained.

160 – Figure 1. Although the image is just an example, which is interesting, a reference of which varieties are the numbers provided above the gel should be provided or added to a results table. It may be interesting for some readers to see the results of the individual responses of some varieties in the gels.

201 – 3.4. Cluster analysis of the 119 rice main varieties … This part is interesting, especially the clustering according to levels of resistance and association with regions/climate? from Jiangsu. However, clustering was done by presence/absence of genes, as expressed in M&M. It would be interesting to relate these clusters with the genes present.

205 - … sub-clusters IA-1 and IA-2. … these are subclusters within the other subclusters.

244 – combination … combinations.

244 – Among them, there were six combinations with 100% resistance … (Table S4) … I understand that in table S4 the combinations with the highest level of resistance (80-100%) are shown, but in the text only those of 100% are named. Interesting that Kendao88 (Table S4) has composite value 5, is this because of the of panicle blast incidence?

271 –  Moreover, The re- … Moreover, the…

283 – … 1,933,000 hm2 … What unit express? Previously was used hectares.

361 – 371 – This part needs to be edited.

Author Response

Review of Molecular detection and analysis of blast resistance genes in main rice varieties in Jiangsu province, China, submitted for possible publication in Agronomy.

Comments and Suggestions for Authors

The article is well done and provides interesting information. In general, this article is well written, but some polishing of the English text is suggested. Some suggestions/comments for improving the manuscript are provided below.

35 – 36 - Rice blast is prevalent in rice-growing areas in Jiangsu province due to the climate transition zone between north and south… What are the climatic differences between north and south that make blast prevalent?

Response: We thank the reviewer for the positive comments. We have modified this section with “The rice-growing areas in Jiangsu province are located in the south-north climate transition zone, which results the suitable temperature and more precipitation for the occurrence of rice blast, and the average annual occurrence area reaches about 11 million mu, which seriously threatens the safe production of rice” (line 36-39).

71 - … Liu et al … add a dot after al. and the reference number. Check that all other references are correctly cited.

Response: Thanks. We have made correction in the full manuscript.

76 - Rice blast can be classified into … this part should go in a separate paragraph.

Response: Thanks. We have made modification.

89 - A set of 119 rice main … main varieties of Jiangsu province?

Response: We agree with the reviewer’s point and we have modified with “A set of 119 rice main varieties (94 Japonica and 25 Indica) were collected from the seed management station of Jiangsu province, and whcih were planted in all the growing regions (Northern, Central and Southern Jiangsu) of Jiangsu province. These varieties have a large planting area and represent the main varieties in Jiangsu Province” (line 92-95)

110 - The spectrum of disease reaction … add that it is for panicle blast. This is severity of panicle blast.

Response: Thanks. We have made modification.

115 – Standard for grading the incidence of rice panicle blast: … What degree of severity was used for incidence evaluation, or was any symptom or degree of severity used to calculate incidence? According with Table S1 100 panicles were evaluated.

Response: The incidence was calculated on the basis of the total number of panicles with level 1-9 of the spectrum of disease reaction for panicle blast in 100 panicles surveyed.

124 - Where does this Composite Index formula come from? If it was created here, it should be explained.

Response: The composite index is an artificial concept that synthesizes the comprehensive resistance of a variety to leaf blast and panicle blast. In order to better understand the calculation of the composite index, we have modification the description with “Standard for grading composite index of rice blast: level 0: <0.1; level 1: 0.1-2.0; level 3: 2.1-4.0; level 5: 4.1-6.0; level 7: 6.1-7.5; level 9: 7.6-9.0. Composite index = level of leaf blast×25%+ level of incidence of panicle blast× 25%+ level of loss index of panicle blast×50%. Take the rice variety Suxianggeng1785 as an example, the composite index=2×0.25+9×0.25+5×0.5=5.25, based on the standard, the grading incidence is level 5.”

160 – Figure 1. Although the image is just an example, which is interesting, a reference of which varieties are the numbers provided above the gel should be provided or added to a results table. It may be interesting for some readers to see the results of the individual responses of some varieties in the gels.

Response: We thank the reviewer for the critical but helpful comment. We have added the variety number in Table S2.

201 – 3.4. Cluster analysis of the 119 rice main varieties … This part is interesting, especially the clustering according to levels of resistance and association with regions/climate? from Jiangsu. However, clustering was done by presence/absence of genes, as expressed in M&M. It would be interesting to relate these clusters with the genes present.

Response: We thank the reviewer for the critical but helpful comment. We analyzed the cluster of the 119 rice main varieties and found that most indica varieties planted in northern Jiangsu were grouped in IB sub-cluster and these indica varieties showed better resistant. In addition, we analyzed the distribution of rice varieties contained R genes, most R genes were relatively consistent regionally, but the rice varieties containing Pi9, Pi5 and Piz-t were mostly grown in northern Jiangsu, while the rice varieties containing Pib, Pi1 and Pia were mostly found in central and southern Jiangsu. We have added the section in the manuscript in line 227-231.

Table S4 The distribution rate of rice varieties containing R genes in three regions of Jiangsu province.

Resistance genes

No. of varieties

No. of varieties (%)

Northern Jiangsu

Central Jiangsu

Southern Jiangsu

Pish

119

74(100%)

21(100%)

24(100%)

Pit

114

70(94.59%)

21(100%)

23(95.83%)

Pib

84

46(62.16%)

17(80.95%)

21(87.50%)

Pi54

44

28(37.84%)

6(28.57%)

10(41.67%)

Pita

53

30(40.54%)

10(47.62%)

13(54.17%)

Pi9

56

40(54.05%)

8(38.10%)

8(33.33%)

Pi2

18

11(14.86%)

3(14.29%)

4(16.67%)

Pi1

84

48(64.86%)

16(76.19%)

20(83.33%)

Pikm

36

23(31.08%)

8(38.10%)

5(20.83%)

Pigm

2

0

1(4.76%)

1(4.17%)

Pia

96

55(74.32%)

19(90.48%)

22(91.67%)

Pi5

29

27(36.49%)

2(9.52%)

0

Pb1

52

31(41.89%)

11(52.38%)

10(41.67%)

Piz-t

29

21(28.38%)

3(14.29%)

5(20.83%)

205 - … sub-clusters IA-1 and IA-2. … these are subclusters within the other subclusters.

Response: Thanks. We have checked this section and found that the varieties contained in the sub-clusters were not duplicated.

244 – combination … combinations.

Response: Thanks. We have made modification.

244 – Among them, there were six combinations with 100% resistance … (Table S4) … I understand that in table S4 the combinations with the highest level of resistance (80-100%) are shown, but in the text only those of 100% are named. Interesting that Kendao88 (Table S4) has composite value 5, is this because of the of panicle blast incidence?

Response: We agree with the reviewer’s point and we have added the gene combination Pita+Pikm+Pi5 in the text. (line 258)

271 –  Moreover, The re- … Moreover, the…

Response: Thanks. We have made modification.

283 – … 1,933,000 hm2 … What unit express? Previously was used hectares.

Response: Thanks. We have made modification.

361 – 371 – This part needs to be edited.

Response: We thank the reviewer for the positive comments and we have edited.